# Stop Reasoning! When Multimodal LLM with Chain-of-Thought Reasoning Meets Adversarial Image

**Zefeng Wang**[*]
Technical University of Munich

**Zhen Han**[*]
LMU Munich

**Shuo Chen**
LMU Munich

**Fan Xue**
Technical University of Munich

**Zifeng Ding**
LMU Munich

**Xun Xiao**
Huawei Technologies

**Volker Tresp**
LMU Munich

**Philip Torr**
University of Oxford

**Jindong Gu**[†]
University of Oxford

## Abstract

Multimodal LLMs (MLLMs) with a great ability of text and image understanding have received great attention. To achieve better reasoning with MLLMs, Chain-of-Thought (CoT) reasoning has been widely explored, which further promotes MLLMs' explainability by giving intermediate reasoning steps. Despite the strong power demonstrated by MLLMs in multimodal reasoning, recent studies show that MLLMs still suffer from adversarial images. This raises the following open questions: Does CoT also enhance the adversarial robustness of MLLMs? What do the intermediate reasoning steps of CoT entail under adversarial attacks? To answer these questions, we first generalize existing attacks to CoT-based inferences by attacking the two main components, i.e., rationale and answer. We find that CoT indeed improves MLLMs' adversarial robustness against the existing attack methods by leveraging the multi-step reasoning process, but not substantially. Based on our findings, we further propose a novel attack method, termed as stop-reasoning attack, that attacks the model while bypassing the CoT reasoning process. Experiments on three MLLMs and two visual reasoning datasets verify the effectiveness of our proposed method. We show that stop-reasoning attack can result in misled predictions and outperform baseline attacks by a significant margin. The code is available here.

## 1 Introduction

Previous research has shown that traditional vision models (e.g. image classifiers) are vulnerable to images with imperceptible perturbations, exposing a significant challenge in AI security (Szegedy et al., 2013; Goodfellow et al., 2014). Recently, multimodal large language models (MLLMs) have demonstrated impressive competence in image understanding with the knowledge learned by LLMs, which arises the interest in studying whether MLLMs also show vulnerability to adversarial images. Some recent works confirm that MLLMs are also vulnerable to adversarial images with significant performance drops (Zhao et al., 2023; Bailey et al., 2023; Luo et al., 2024), showing the importance in studying the adversarial robustness of MLLMs.

To improve MLLM's performance in understanding images with complex content, Chain-of-Thought (CoT) reasoning has been explored in MLLMs (Lu et al., 2022; Zhang et al., 2023; He et al., 2023). CoT reasoning generates intermediate reasoning steps, known as rationale, before predicting the answer. This approach not only improves models' inference

---

[*]Equal contribution.
[†]Corresponding author: jindong.gu@outlook.com

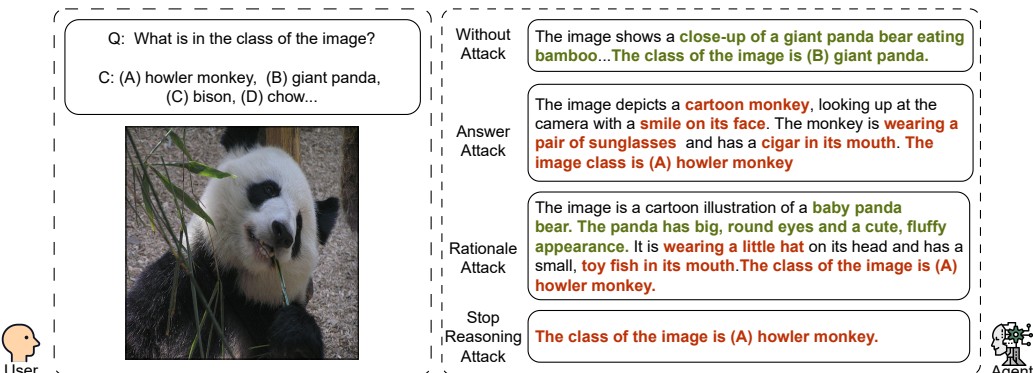

Figure 1: Given adversarial images, *answer attack* and *rationale attack* make an MLLM output an **explanation for the incorrect predictions** with CoT . The phrases highlighted with red are found to inaccurately depict the actual facts. Apart from these two attacks, *stop-reasoning attack* shows the ability to **restrain the reasoning process** and make an MLLM output an incorrect answer even if the model is prompted to leverage the CoT explicitly.

power but also introduces explainability, which is essential in critical domains such as clinical decision-making (Singhal et al., 2022). Nevertheless, the performance of CoT-based inference in MLLMs when facing adversarial images is still not fully investigated. In this work, we primarily explore the following questions:

- Does CoT enhance the adversarial robustness of MLLM?
- What do the intermediate reasoning steps of CoT entail under adversarial attacks?

Since CoT-based inference consists of two parts, i.e., rationale and final answer, we investigate the adversarial robustness of MLLMs by attacking both of them. First, two existing attacks are generalized to MLLMs with CoT, i.e., *answer attack* and *rationale attack*. *Answer attack* attacks only the extracted choice letter in the answer, e.g., the "B" character in Figure 1, which is suitable for both MLLMs with or without CoT. The other attack, *rationale attack*, attacks not only the choice letter in the answer but also the preceding rationale (Figure 1 *rationale attack*). We find that models employing CoT tend to demonstrate considerably higher robustness under both *answer* and *rationale attacks* compared with models without CoT.

Based on this observation, we further devise a new attacking method called *stop-reasoning attack*. *Stop-reasoning attack* aims to interrupt the reasoning process and force the model to directly answer the question even with an explicit requirement of CoT in the prompt. Meanwhile, the choice letter is also attacked, leading the model to predict an incorrect answer (as shown in Figure 1 *stop-reasoning attack*). In this way, the enhancement brought by CoT is limited, making MLLMs more vulnerable to adversarial attacks.

Furthermore, with the existing two attacks, i.e., *answer attack* and *rationale attack*, the CoT mechanism elucidates the model's intermediate reasoning steps, which opens a window for us to understand the reason for an incorrect answer when encountering adversarial images. As shown in Figure 1, with *answer attack*, even though only the choice is attacked ("B" → "A"), the rationale changes correspondingly and reveals the reason: the panda with black eyes is misidentified as a monkey with sunglasses. For *rationale attack*, although the panda is correctly recognized in the rationale, the other wrong information in the rationale influences the answer and leads to a wrong answer.

We conduct experiments with MiniGPT4 (Zhu et al., 2023), OpenFlamingo (Awadalla et al., 2023), and LLaVA (Liu et al., 2023a) as the representatives of victim MLLMs on two visual question answering datasets that require understanding on complex images, i.e., A-OKVQA (Schwenk et al., 2022) and ScienceQA (Lu et al., 2022). Experimental results demonstrate that MLLMs with CoT exhibit enhanced robustness compared to MLLMs

without CoT across diverse datasets. Our *stop-reasoning attack* can restrain the CoT reasoning process even with the explicit prompt requiring CoT. It leads to a higher success rate, results in misled predictions, and outperforms baselines by a significant margin.

To summarize, we have the following contributions:

- We study the influence of CoT on the adversarial robustness of MLLMs by performing attacks on the two core components of CoT, i.e., *rational* and *answer*.

- We propose a novel attack method, i.e., *stop-reasoning attack*, for MLLMs with CoT, which is effective at the most.

- We show that the rationale opens a window for understanding the reason for an incorrect answer with an adversarial image.

- Extensive experiments are conducted on representative MLLMs and two datasets under the proposed attacking methods to justify our proposal.

## 2 Related work

### 2.1 Adversarial attacks

Deep learning models are known to be vulnerable to adversarial attacks (Szegedy et al., 2013; Goodfellow et al., 2014). Extensive previous studies have a primary focus on image recognition (Szegedy et al., 2013; Goodfellow et al., 2014; Athalye et al., 2018; Carlini & Wagner, 2017; Gu et al., 2021) and many well-known adversarial methods are proposed such as Projected Gradient Descent (PGD) (Madry et al., 2017), Fast Gradient Sign Method(FGSM) (Goodfellow et al., 2014). These studies aim to mislead the models to generate wrong predictions while only adding minimal and imperceptible perturbations to the images (Goodfellow et al., 2014). Unicorn (Tu et al., 2023) explores robustness against adversarial attacks and OOD generalization but does not address the influence of CoT reasoning on MLLMs' robustness. AVIBench (Zhang et al., 2024) and MM-SafetyBench (Liu et al., 2023b) provide frameworks for evaluating MLLMs' robustness against adversarial attacks, but they do not focus on the CoT reasoning process. Despite the effectiveness of these attacks, it is still hard to interpret the model behavior during the attacks and understand why the attacks could succeed (Gu & Tresp, 2019; Li et al., 2022). Recent studies have also investigated the vulnerability of large language models (Zou et al., 2023; Kumar et al., 2023) and multimodal LLMs (Zhao et al., 2023; Gan et al., 2020; Gao et al., 2024; Han et al., 2023) under adversarial attacks. However, the adversarial robustness of multimodal LLMs with CoT reasoning ability is still under-explored. Since CoT reasoning reveals the model's decision process (Wei et al., 2023), this reported intermediate process can serve as a good proxy for to understand the model behavior before and after the adversarial attacks, which additionally brings explainability. Different from previous studies, this work focuses on evaluating the adversarial robustness of MLLMs with CoT by designing effective attack methods and understanding why the model would behave under such adversarial attacks.

### 2.2 Chain-of-thought reasoning on multimodal LLMs

CoT generates a series of intermediate logical reasoning steps and assists LLMs in thinking step by step before generating the final answer (Wei et al., 2023). CoT has been widely applied to LLMs (Wei et al., 2023; Kojima et al., 2023; Zhang et al., 2022) and has significantly improved the performance in various tasks, such as arithmetic problems (Wei et al., 2023) and symbolic reasoning (Wei et al., 2023). Some studies have noticed that CoT can bring extra robustness to the LLMs (Wu et al., 2023) and have designed a better CoT method for better robustness (Wang et al., 2022) Recently, on MLLMs, various studies have also shown that adopting CoT on MLLMs can bring superior performances as well (Lu et al., 2022; Zhang et al., 2023; He et al., 2023). However, the robustness of CoT on MLLMs against adversarial attacks has not been investigated. It is still an open question whether CoTreasoning is beneficial, indifferent, or even harmful to the robustness of MLLMs under adversarial attacks. This study aims to first evaluate the adversarial robustness of CoT on MLLMs and then understand how the attacks affect the model behavior.

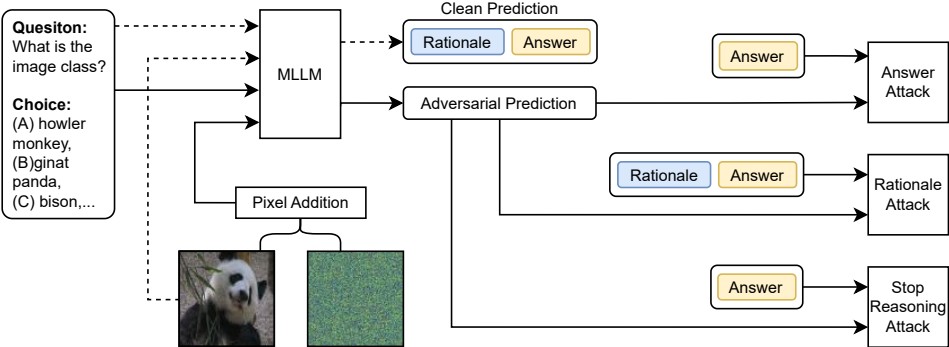

Figure 2: Pipeline. The dotted line indicates a clean prediction with the original image. The solid line visualizes the attack pipeline in one iteration. The adversarial image $v_{adv}$ is built with the corresponding attack method.

## 3 Methodology

### 3.1 Threat models

This work examines the influence of the CoT reasoning process on MLLMs' adversarial robustness. We follow the principles introduced by Carlini et al. (2019) to define our adversary goals, adversarial capabilities, and adversary knowledge. The **adversary goal** is to cause the model to output a wrong answer. Given the scenario that MLLMs with a prompt are applied to extract information from user images, the images are assumed to be manipulated by an attacker to mislead MLLMs. Hence, we restrict the **adversarial capability** to perturb the image in an imperceptible range and assume text prompts are unmodifiable. The restrictions on images are

$$\mathcal{D}(v_{org}, v_{adv}) = \max \left| v_{org} - v_{adv} \right| \leq \epsilon \tag{1}$$

where $\mathcal{D}(\cdot)$ is the distance between images, $v_{org}$ is the original input image, $v_{adv}$ is the perturbed image, and $\epsilon$ is a predefined boundary. As for the **adversary knowledge**, we assume the full knowledge of the model. Thus, solid attacks can be performed with the PGD (Madry et al., 2017) method for convincing results.

### 3.2 Attack pipeline

We denote a visual question answering (VQA) inference as $f(v, q) \mapsto t$, where $f(\cdot)$ represents an MLLM, $v$ is the input image, $q$ is input text formulated as a question with its multiple answer choices, and $t$ is the output of the MLLM. To make models use the CoT reasoning process, we add a prompt as explicit instruction after the question and choices, e.g., "First, generate a rationale with at least three sentences that can be used to infer the answer to the question. At last, infer the answer according to the question, the image, and the rationale.".

As depicted in Figure 2 dotted line, both the textual question and corresponding image are fed to an MLLM to produce an initial clean prediction. This clean prediction, denoted as $t_{clean}$, serves as the basis for calculating losses according to three attack methods.

In one attack iteration (Figure 2 solid line), the MLLM takes both the perturbed image from the last attack iteration $v_{adv_{last}}$ and text $q$ as input and generates an adversarial output. Then, the new adversarial image $v_{adv_{new}}$ is built by leveraging different attack methods. In the first attack iteration, an initial perturbation is performed before the image is fed into the model. The corresponding optimization problem can be defined as:

$$\underset{\mathcal{D}(v_{org}, v_{adv}) \leq \epsilon}{\arg \max} \quad \mathcal{L}(f(v_{adv}, q), f(v_{org}, q)) \tag{2}$$

the optimization problem can be solved with the PGD method.

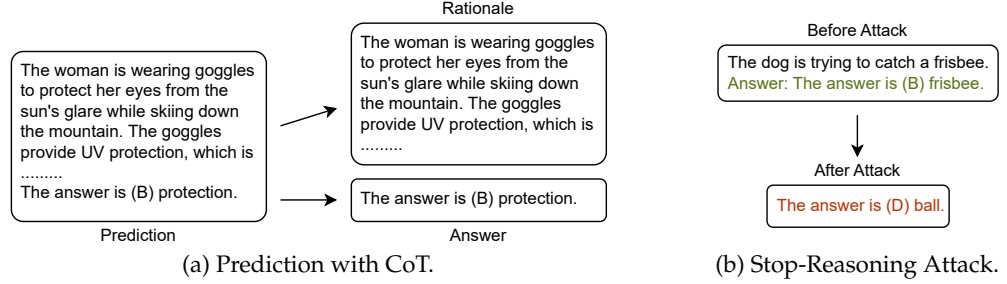

(a) Prediction with CoT.        (b) Stop-Reasoning Attack.

Figure 3: Models output rationale and answer as prediction without attack. After *stop-reasoning attack*, models output only the answer. (a) Prediction with CoT. The complete prediction with CoT can be divided into two components: the rationale and the answer. (b) Stop-Reasoning Attack. After *stop-reasoning attack*, MLLMs skip the reasoning part and output the answer directly without rationale.

## 3.3 Generalized attacks

As shown in Figure 3a, model inference with CoT reasoning provides an answer and a rationale as its prediction output. We first generalize two existing attack methods to MLLMs with CoT, i.e. *answer attack* and *rationale attack*

**Answer Attack.** The answer attack focuses exclusively on attacking the answer part of the output, aiming to manipulate the model to infer a wrong answer.

To alter the answer in the prediction, a cross-entropy loss is computed between the generated answer and the ground truth. We extract the explicit answer choice to ensure that the loss computation focuses solely on the chosen response (please refer to Appendix B.2) Given the loss depends only on one character, the influence of the prediction's length is mitigated. The loss function is defined as follows:

$$\mathcal{L}_{ans}(t_{adv}, t_{clean}) = CE(g(t_{adv}), g(t_{clean})) \tag{3}$$

where $g(\cdot)$ is the answer extraction function, $CE$ is the cross-entropy function. With escalating the loss, models infer alternative answers, deviating from the correct responses.

**Rationale Attack.** Upon revealing the inferences of models with CoT under *answer attack*, an interesting observation surfaced: despite the disregard for the rationale in the attack's design, the rationale part also changes in most cases. Building on this insight, the *rationale attack* is performed, which, in addition to targeting the answer part, also aims at modifying the rationale. We utilize the Kullback-Leibler (KL) divergence to induce changes in the rationale because a high KL divergence indicates a high information loss that fits the target, making the rationale more useless, while cross-entropy does not fit to measure the information entropy. Specifically, the loss function of *rationale attack* is as follows

$$\mathcal{L}_{rsn}(t_{adv}, t_{clean}) = KL(t^{rat}_{adv}, t^{rat}_{clean}) + \mathcal{L}_{ans}(t_{adv}, t_{clean}) \tag{4}$$

where the $t^{rat}_{adv}$ is the rationale in the adversarial output and the $t^{rat}_{clean}$ is the rationale in the clean prediction. As the KL divergence increases by perturbing the image, the adversarial rationale diverges from the clean rationale. Hence, an alternative answer is predicted based on the altered rationale.

The results indicate that CoT slightly boosts the adversarial robustness of MLLMs against the aforementioned two existing methods and introduces the explainability of the incorrectness.

## 3.4 Stop-reasoning attack

Having explored the influence of CoT on the adversarial robustness of MLLMs, we found that the rationale is important for the inference process. A pertinent question arises: how will the model behave when the reasoning process is halted? Inspired by this question, we introduce *stop-reasoning attack*, a method that targets the rationale to interrupt the model's

| Model | Dataset | w/o CoT | | with CoT | | |
|---|---|---|---|---|---|---|
| | | w/o Attack | Answer Attack | Answer Attack | Rationale Attack | Stop Reasoning Attack |
| MiniGPT4-7B | A-OKVQA | 61.38 | 0.76 | 16.06 | 29.06 | **2.87** |
| | ScienceQA | 66.28 | 1.17 | 31.51 | 44.40 | **11.20** |
| MiniGPT4-13B | A-OKVQA | 42.65 | 1.45 | 17.97 | 36.84 | **10.53** |
| | ScienceQA | 63.64 | 13.75 | 33.78 | 45.69 | **30.89** |
| Open-Flamingo | A-OKVQA | 34.80 | 3.52 | 11.14 | 10.79 | **4.95** |
| | ScienceQA | 34.55 | 3.66 | 34.73 | 28.87 | **20.04** |
| LLaVA | A-OKVQA | 92.21 | 0.74 | 36.22 | 21.88 | **12.02** |
| | ScienceQA | 83.17 | 1.13 | 56.96 | 49.27 | **22.39** |

Table 1: Inference accuracy (%) results of victim models. The samples achieve 100% accuracy with models employing the CoT reasoning process. Across diverse attacks, when models are prompted with CoT, *stop-reasoning attack* emerges as the most effective method. For further studies and the experiment with ImageNet dataset, please refer to Appendix D

reasoning process. The objective of this attack is to compel the model to predict a wrong answer directly without engaging in the intermediate reasoning process.

In the text input, we predefined a specific answer template, denoted as $t_{tar}$, to prompt the model to output the answer in a uniform format. The upper part of Figure 3b shows that well-finetuned MLLMs are able to produce answers following the prompt. Therefore, when the initial tokens align with the answer format $t_{tar}$, the model is forced to directly output the answers in the predefined format and bypass the reasoning process even if the model is prompted explicitly to leverage the CoT (refer to the lower part of Figure 3b).

*Stop-reasoning attack* formulates a cross-entropy loss to drive the model towards inferring the answer directly without a reasoning process:

$$\mathcal{L}_{stop}(t_{adv}, t_{clean}, t_{tar}) = -CE(t_{adv}, t_{tar}) + \mathcal{L}_{ans}(t_{adv}, t_{clean}) \tag{5}$$

where $t_{tar}$ is a predefined answer template, e.g., "*The answer is* ().$[EOS]$". By increasing the loss, MLLMs directly output the answer by aligning the initial tokens with the specified answer format and alter the answer into a wrong one. This approach bypasses the reasoning process and thus, it eliminates the robustness boost introduced by CoT. The results on all models and datasets reveal its effectiveness. *Stop-reasoning attack* outperforms the two existing methods by a large margin and can be close to the results of MLLMs without CoT.

## 4 Experiments

### 4.1 Experimental settings

**Datasets.** ScienceQA (Lu et al., 2022) and A-OKVQA (Schwenk et al., 2022) are used to evaluate the impact of the CoT reasoning process on the adversarial robustness of MLLMs, where both datasets comprise multiple-choice questions and rationales and require understanding on complex images. ScienceQA is sourced from elementary and high school science curricula and includes reasoning tasks. A-OKVQA requires commonsense reasoning about the depicted scene in the image and is known as a prevalent choice for VQA reasoning tasks. We perform attacks on data samples that are correctly answered by MLLMs with CoT (need to be attacked (correctly answered) and can be attacked with *rationale* and *stop-reasoning attacks* (with CoT)).

**Victim Models.** Three representative MLLMs are used in our experiments, *i.e.*, MiniGPT4 (Zhu et al., 2023), OpenFlamingo (Awadalla et al., 2023) and LLaVA (Liu et al., 2023a). Commercial MLLMs like GPT-4 (OpenAI et al., 2023), which operate as black-box products, are excluded from the experiments because the first-order gradients for perturbation are inaccessible. Note that MiniGPT4 and OpenFlamingo can infer with CoT without fine-tuning, while LLaVA initially lacks the CoT capability. LLaVA acquires the

CoT capability through fine-tuning with CEQA (Bai et al., 2023). In our work, we experiment on MiniGPT4-7B, OpenFlamingo-9B, and LLaVA-1.5-7B. For detailed parameters and experiment settings, please refer to Appendix A.

### 4.2 How does CoT influence the robustness of MLLMs?

In this section, we present the evaluation results of the three victim models under the three proposed attacks to answer the following two questions:

- *Does the CoT reasoning bring extra robustness to MLLMs against adversarial images?* From the results of *answer attack* and *rationale attack* in Table 1, the CoT brings a marginal robustness boost against the two existing attacks.

- *Is there any specific attack targeting MLLMs with CoT that is effective?* Comparisons in Table 1 show that *stop-reasoning attack* is the most effective for MLLMs with CoT.

We provide more diverse examples in Appendix F, where the consequences caused by the three attacks are illustrated.

### 4.2.1 CoT marginally enhances robustness only on existing attacks

As shown in Table 1, without CoT, the considered models exhibit high vulnerability. Under *answer attack*, on the A-OKVQA dataset, the accuracy of MiniGPT4 without CoT drops to 0.76%, while its accuracy can still remain at 16.06% with CoT. Similarly, on the ScienceQA dataset, the accuracy of MiniGPT4 drops to 1.17% when answering without CoT under *answer attack*, while if with CoT, it can remain at 31.51%. We observe the same trends for the ScienceQA and A-OKVQA datasets on OpenFlamingo and LLaVA and find that models' robustness is relatively boosted by using CoT.

|  | MiniGPT4 | OpenFlamingo | LLaVA |
|---|---|---|---|
| Changed | 100 | 84.25 | 97.89 |
| Not Changed | 0 | 15.75 | 2.11 |

Table 2: Distribution (%) of rationale changes. When *answer attack* succeeds on MLLMs with CoT, although *answer attack* specifically targets on the final answer, a majority of samples exhibit altered rationales.

After reviewing the examples, an important observation is noted that the majority of samples suffering successful *answer attacks* exhibit altered rationales, even though *answer attack* does not aim at the rationale part (Table 2). This implies that attacking a model with CoT requires changing both the answer and rationale parts.

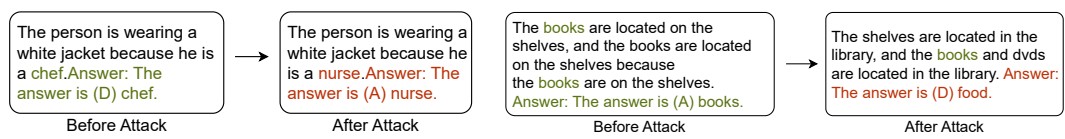

(a) Key change visualization.  (b) Trivial change visualization.

Figure 4: Rationale with key changes and trivial changes after attacks. (a) Key change visualization. The replication of the answer serves as the key information to infer the answer from the rationale. After *answer attack*, the keyword in the rationale is also altered, even though the attack exclusively targets on the answer ("D" → "A"). (b) Trivial change visualization. The replication of the answer is the key information to infer the answer from the rationale. After *answer attack*, the keyword is not changed (the word "books" is not changed), while the other part of the rationale is changed.

Based on the observation above, *rationale attack* is performed. *Rationale attack* exhibits superior performance on OpenFlamingo and LLaVA compared to *answer attack* (Table 1),

with marginal improvement (56.96% to 49.27% on ScinceQA on LLaVA, 11.14% to 10.79% on A-OKVQA on OpenFlamingo). Conversely, on MiniGPT4, the rationale attack proves less effective than *answer attack* on both datasets (16.06% under *answer attack* against 29.06% under *rationale attack*).

To understand why *rationale attack* does not always work, we pick 100 samples of each victim model on A-OKVQA under *rationale attack*. We classify these samples into two categories according to their changes in rationale: key changes and trivial changes. A key change refers to the modifications on words crucial for deducing a correct answer, as shown in Figure 4a. A trivial change (as illustrated with the example in Figure 4b), on the other hand, refers to those modifications on words that are non-crucial for deducing a correct answer while leaving key information untouched.

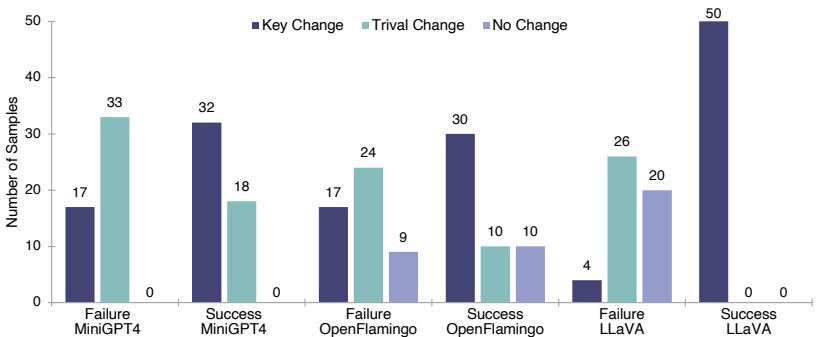

Figure 5: Classifications of different types of changes made to rationale in three victim models under *rational attack* (based on 100 Samples/Model). The groups "Failure" and "Success" indicate whether the attack failed or succeeded. "Failure" indicates an unsuccessful attack where the model's prediction remains correct, while "Success" denotes a successful attack resulting in a change from a correct to an incorrect prediction.

Figure 5 gives statistical comparisons of the respective numbers of different types of changes made via *rationale attack* to the three victim models. The classifications indicate that successfully attacked samples under *rationale attack* are often associated with significant modifications to key information within the rationale. Conversely, samples lacking altered rationales or featuring only minor adjustments tend to preserve their correct answers. This suggests the critical role of key information in influencing the inference of the final answer. However, precisely identifying the crucial information is hard, and modifying it efficiently is even more difficult. To this end, *rationale attack* can only slightly enhance the attack performance in comparison with *answer attack*, and our results further support the finding that CoT improves adversarial robustness against generalized attack methods.

### 4.2.2 Stop-reasoning attack's effectiveness

Given the ineffectiveness of the answer and *rationale attacks*, we introduce *stop-reasoning attack* to halt the model's reasoning. The results demonstrate that *stop-reasoning attack* outperforms both other attacks (11.20% against 31.51% and 44.40% on SincecQA on MiniGPT4, 4.95% against 11.14% and 10.79% on A-OKVQA on Open-Flamingo, and 12.02% against 36.22% and 21.88% on A-OKVQA on LLaVA). It even approaches the performance observed when attacking models without CoT (2.87% against 0.76% on A-OKVQA on MiniGPT4), indicating its remarkable potency in mitigating the additional robustness introduced by the CoT reasoning process. Figure 6 illustrates an example where both *rationale* and *answer attacks* fail, and only *stop-reasoning attack* succeeds. In this work, simple outcheck defense does not work, resulting high false positive rate, given the fact that a simple image does not require CoT.

To understand the effectiveness of stop-reasoning, we examine the results of *stop-reasoning attack* and observe that after the attack, the model outputs the answer directly without

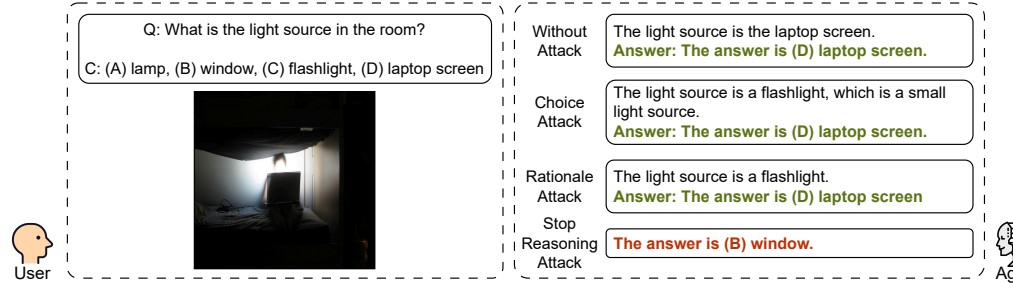

Figure 6: Example of all attacks. *Stop-reasoning attack* is potent. At the top, it shows the callouts of a user with an input image and their associated textual questions. The four callouts below are the answers from the MLLM under each type of attack. Only *stop-reasoning attack* achieves the goal of failing the model by providing a wrong answer (highlighted in red color).

leveraging CoT, aligning with the fundamental concept of *stop-reasoning attack* – aiming to halt the CoT reasoning process (Figure 6 and more examples in Appendix F). When *stop-reasoning attack* succeeds, the model disregards the prompt's CoT reasoning process requirement and directly infers the answer.

As revealed in Section 4.2.1, the extra robustness boost is intricately tied to the generated rationale. If this CoT reasoning process is halted by *stop-reasoning attack*, the additional robustness generated during the CoT reasoning process will be diminished as well. Thus, achieving the adversary's goal becomes comparatively easier.

Concerning the performance gap between the *stop-reasoning attack* on MLLMs with CoT and the *answer attack* on MLLMs without CoT. The difference might be caused by two factors: Firstly, the prompts for CoT and non-CoT scenarios are naturally different. Explicit instructions are added to prompt the MLLMs to leverage the CoT. In comparison, the MLLMs are prompted to infer the answer directly under "w/o CoT" scenarios. Secondly, the *stop-reasoning attack* cannot always stop MLLMs from performing reasoning. We observed that not all samples successfully skipped the reasoning phase after the *stop-reasoning attack* (e.g., when evaluating the *stop-reasoning attack* with the MiniGPT4 model on the A-OKVQA dataset, the model did not skip the reasoning phase for 5.35% samples).

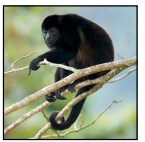

The image depicts a **giant panda sitting on a tree branch**, looking downwards. The panda has **black fur with white patches on its ears, face, and chest.** It has a long, curved claw on each paw.
The class of the image is **giant panda**.

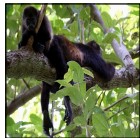

The image depicts a **giant panda sitting on a tree branch** in a lush green forest. The panda has black and white fur, with a **distinctive black patch on its forehead.**
The class of the image is **giant panda.**

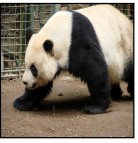

The image is a **black and white photograph of a bison**. The bison is large and has a shaggy coat with a long mane. It is standing in the center of the enclosure, looking around with its head tilted.
The class of the image is **bison**.

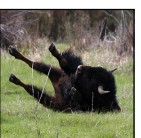

The image shows a **black dog** lying on its back in a grassy field. The dog is **wearing a collar** and has its paws in the air. There are trees and mountains in the background.
The class of the image is **chow**.

(a) CoT brings explainability for answer attack.  (b) CoT brings explainability for rationale attack.

Figure 7: Comparison of CoT under different attacks. (a) CoT brings explainability under *answer attack*. On the top, a monkey is falsely recognized as a panda. On the bottom, a panda is falsely recognized as a bison. (b) CoT brings explainability under *rationale attack*. On the top, a monkey is falsely recognized as a panda. On the bottom, a bison is falsely recognized as a chow.

### 4.3 What does the rationale entail under adversarial attacks?

Although CoT brings marginal robustness to MLLMs against existing attacks, MLLMs are still vulnerable to adversarial images, similar to traditional vision models. When traditional vision models make inferences, e.g., on classification tasks, our understanding is confined to the correctness of the answer. Delving deeper into the model's reasoning process and answering the question of why the model infers a wrong answer with an adversarial image is difficult. In comparison, when MLLMs perform inference with CoT reasoning, it opens a window into the intermediate reasoning steps that models employ to derive the final answer. The intermediate reasoning steps (rationale part) generated by the CoT reasoning process provide insights and potentially reveal the reasoning process of the MLLMs.

To look deeper into the explainability introduced by CoT, we conducted image classification tasks on ImageNet (Russakovsky et al., 2015). These tasks involved constructing multi-choice questions by extracting subsets from ImageNet (please refer to Appendix C for selected classes). We provide two example pairs to illustrate the rationale's changes under *answer attack* and *rationale attack*.

Figure 7a illustrates CoT inference under *answer attack*. In the upper example, CoT erroneously interprets the partial color of the monkey as white, resulting in the misclassification of the monkey as a panda. In the bottom example, the rationale falsely asserts that the black-and-white patterns on the panda's body resemble the black-white picture of a bison. This misconception leads to the incorrect inference of a bison. Figure 7b displays examples under *rationale attack*. In the upper example, the rationale incorrectly states that the black forehead is a distinctive black patch, leading the model to inaccurately classify the image as a panda instead of a monkey. In the bottom example, the horn of a bison is misinterpreted as a collar, resulting in the false classification of the bison as a chow.

## 5 Conclusion

In this paper, we fully investigated the impact of CoT on the robustness of MLLMs. Specifically, we introduced *stop-reasoning attack*, a novel attack method tailored for MLLMs with CoT. Our findings reveal that CoT can slightly enhance the robustness of MLLMs against *answer attack* and *rationale attack*. This extra robustness is attributed to the complexity of changing precisely the key information in the rationale part. For *stop-reasoning attack*, our test results show that MLLMs with CoT still suffer and the expected extra robustness is eliminated. At last, examples are provided to reveal the changes in CoT when MLLMs infer wrong answers with adversarial images.

**Acknowledgements**

This paper is supported by the DAAD programme Konrad Zuse Schools of Excellence in Artificial Intelligence, sponsored by the Federal Ministry of Education and Research.

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

# Appendix

## A Attack settings

Across all attack scenarios, the perturbation constraint $\epsilon$ is set to 16. The maximum number of attack iterations is capped at 200. To generate the adversarial output $t_{adv}$, we opt for the $forward(\cdot)$ function over the $generate(\cdot)$ function in MLLMs. This choice is driven by the fact that the $generate(\cdot)$ function demands significantly much more time, rendering the attack impractical due to prolonged running times across extensive iterations. The prediction is updated every 10 iterations to mitigate the gap between the $forward(\cdot)$ method and the $generate(\cdot)$ method. In every attack test, all victim models use a 0-shot prompt to output their final answer. Every attack method starts with a random perturbation on the image in the very first iteration, then follows its individual loss function and uses PGD method to generate a new perturbed image for the next iteration. The attacks are performed on a single NVIDIA 40G A100 GPU or a single NVIDIA 80G A100 GPU for 13B model. To measure the robustness of the MLLMs, we employ *accuracy* as the performance metric. Low accuracy indicates a low robustness.

## B Implementation details

### B.1 Pseudo code

The algorithm for the entire pipeline is outlined in Algorithm 1. In this algorithm:

- $f_{gen}(\cdot)$ represents the model's inference using the $generate(\cdot)$ method.

- $f_{fw}(\cdot)$ signifies the model's inference using the $forward(\cdot)$ method.

- $D$ is the perturbation constraint.

- The initial adversarial image is created by introducing Gaussian noise to the original image.

- Regular updates to the prediction are essential to alleviate the performance gap between the $forward(\cdot)$ and $generate(\cdot)$ methods.

---

**Algorithm 1** Pipeline

---

**Require:** original image $v_{org}$, question $q$, boundary $\epsilon$, step $\alpha$, maximum iteration $n$
    prediction $t_{clean} = f_{gen}(v_{org}, q)$
    initial adversarial image $v_{adv}$
    truncate adversarial image to fit $\mathcal{D}(v_{org}, v_{adv}) \leq \epsilon$
    **for** $i = 1$ **to** $n - 1$ **do**
        adversarial output $t_{adv} = f_{fw}(v_{org}, q, t_{pred})$
        loss calculation with $\mathcal{L}(t_{adv}, t_{pred})$
        *grad* of $v_{adv}$ from *loss*
        new adversarial image $v_{adv} = v_{adv} + \alpha * sign(grad)$
        check and truncate $\mathcal{D}(v_{org}, v_{adv}) \leq \epsilon$
        **if** update prediction is *true* **then**
            prediction $t_{clean} = f_{gen}(v_{org}, q)$
        **end if**
        **if** *stop criteria* satisfied **then**
            **break**
        **end if**
    **end for**
    $t_{adv} = f_{gen}(v_{adv}, q)$
    save $v_{adv}$
    **return** $t_{adv}$

---

### B.2 Extract answer

To perform an exact attack, the model is prompted to answer the multiple-choice questions in a specific form and explicitly show the answer choice. As shown in Figure 8 (a), only the choice letter in the answer sentence will be considered as the answer. The choice content and choices appearing in other sentences will not be accepted.


<div>
The man in ... The image does not show any signs of sun or snow, which eliminates options (C) and (D). Therefore, the answer is (A) hail.
</div>
<div>
**The image shows a street with a building on the left side and a bicycle parked in front of it. There are no trucks or airplanes visible in the image.** Therefore, the answer is (C) bicycle....
</div>
</div>

(a) Extract answer.                    (b) Split rationale.

Figure 8: (a) Extract answer. Only the choice letter (green) in the answer sentence will be considered as the answer. Other choice letters or choice content (red) will be ignored. (b) Split rationale. Only the sentences (bold) before the answer extracted (green) will be contoured as the rationale.

### B.3 Split rationale and answer

To perform the rationale attack, the rationale and answer parts in the output logit matrix should be split if the model answers the question with the CoT process. As the used LLMs are all generative models, it is not deterministic where the rationale is, where the answer is, and how long each is. So, the output logit matrix is decoded and split into sub-sentences first. As the model is imperfect and the instruction prompt is not strong enough, the model may not follow the instructions exactly. The inference may mix the answer and the rationale part. The part before the sentence the answer is extracted from is the rationale. As shown in Figure 8 (b), inferences can be roughly divided into two parts from the answer. The sentences from the answer are regarded as the answer part of the model, even though there are some other sentences. The sentences before the answer belong to the rationale part. The corresponding logit matrix will be extracted.

### B.4 Stop criteria

The general stop criteria shared in all attack scenarios is whether the inferred answer is wrong in the perturbation loop. The perturbation process will be stopped if the answer is wrong. Then, the perturbed image will be fed into the model again to infer the final answer with the $generate(\cdot)$ method. The sample won't be attacked again, regardless of the correctness of the final answer generated with the $generate(\cdot)$ method. When the MLLM is under *stop-reasoning attack*, the stop criteria is combined with an additional stop check, which checks if the answer is extracted from the first sentence.

### B.5 Forward vs. Generate

In the context of language models, the $forward(\cdot)$ function often refers to the process of passing input data through the model to obtain predictions or activations. For LLMs used in MLLMs like Llama2 (Touvron et al., 2023), the $forward(\cdot)$ function has the same length in output as the input. The output token is the predicted next token to the input token at the same position. The $generate(\cdot)$ function generates output by iteratively using the $forward(\cdot)$ function. Specifically, in each iteration, only the last token, the next token to the whole input, is extracted and concatenated after the input sequence. The new sequence will be used as input in the next iteration until there is an end-of-sequence token $[EOS]$. To generate an output, the $generate(\cdot)$ function costs much more time than the $forward(\cdot)$, if a ground truth can be provided to the $froward(\cdot)$ function, because the $generate(\cdot)$ function goes through the $forward(\cdot)$ function several times while the $forward(\cdot)$ with ground truth needs only one iteration. However, it's important to note that to generate all tokens at once, the $forward(\cdot)$ method requires a ground truth, and there is a performance gap between the $forward(\cdot)$ and the $generate(\cdot)$ functions.

As outlined in appendix B.1, we adopt the clean prediction as the pseudo ground truth for the $forward(\cdot)$ method. As the perturbation progresses, the input image differs from the one used for the clean prediction. Consequently, the pseudo ground truth deviates from the actual ground truth, leading to a divergence in the adversarial output. To address the disparity between the real adversarial output and the actual adversarial output, the clean prediction should be updated with the latest adversarial image every several iterations.

## C   ImageNet subclasses

We create classification tasks by extracting 4, 8, and 16 classes. All the classes are randomly picked from the dataset. The specific classes selected for each scenario are as follows:

- **4 classes**: English setter, Persian cat, school bus, pineapple.
- **8 classes**: bison, howler monkey, hippopotamus, chow, giant panda, American Staffordshire terrier, Shetland sheepdog, Great Pyrenees.
- **16 classes**: piggy bank, street sign, bell cote, fountain pen, Windsor tie, volleyball, overskirt, sarong, purse, bolo tie, bib, parachute, sleeping bag, television, swimming trunks, measuring cup.

All tasks had a uniform question: "What is the class of the image?"

## D   Further studies

### D.1   What If CoT is not necessary for tasks?

We randomly picked several classes from the ImageNet dataset (Appendix C). Surprisingly, as indicated in Table 3, the tests with 8 classes show worse performances than the tests with 16 classes. The reason maybe some of the chosen classes are similar, and the similarity makes the classification task more complex, e.g., American Staffordshire terrier and Shetland sheepdog look similar. However, the conclusion in all tests is consistent: the marginal improvement in performance brought about by CoT suggests that the rationale may not be essential for simple tasks. Although the two main results outlined in Section 4.2.1 share commonalities, a notable gap exists between the accuracy values of the ImageNet series and those of the A-OKVQA and ScienceQA datasets when models with CoT are subjected to the answer attack. This discrepancy can be attributed to the inherent complexity of VQA tasks compared to the straightforward classification tasks on the ImageNet dataset.

Further examination of the A-OKVQA and ScienceQA datasets reveals that the A-OKVQA dataset is relatively easier, as illustrated in Table 1. This performance difference is consistent across all three models. By comparing the accuracy of the classification task on ImageNet with the VQA tasks on A-OKVQA and ScienceQA, a significant observation emerges: CoT has almost no impact on the robustness of simple tasks.

| # Classes | w/o CoT | | with CoT | | |
|---|---|---|---|---|---|
| | w/o Attack | Answer Attack | Answer Attack | Rationale Attack | Stop Reasoning Attack |
| 4 | 85.34 | 0.00 | 4.19 | 4.71 | **0.52** |
| 8 | 82.04 | 0.00 | 1.06 | **0.00** | **0.00** |
| 16 | 74.32 | 0.00 | 3.32 | 2.42 | **0.30** |

Table 3: Inference accuracy (%) on ImageNet classification task. All samples are correctly inferred when inferring with CoT. # classes signifies the number of classes extracted from the ImageNet dataset for multi-choice classification tasks. Figure 9b reveals larger $\epsilon$ values lead to earlier convergence and better performance, but performance plateaus beyond a certain threshold.

### D.2    Ablation study on adversarial capability

During the ablation study, the adversarial capability is changed by narrowing or expanding the limited boundary ($\epsilon$) to 2, 4, 8, 32 (as described in Section 3.1). Table 4 presents results of the MiniGPT4 and Open-Falmingo with $\epsilon$ set to 8. The results are consistent with Table 1, indicating that the CoT reasoning process enhances the robustness of MLLMs. Furthermore, Figure 9 shows the results of ScienceQA on MiniGPT4 with different $\epsilon$. Figure 9a shows *stop-reasoning attack* consistently outperforms *rationale* and *answer* attacks as $\epsilon$ increases.

| Model | Dataset | w/o CoT | | with CoT | | |
|---|---|---|---|---|---|---|
| | | w/o Attack | Answer Attack | Answer Attack | Rationale Attack | Stop Reasoning Attack |
| MiniGPT4 | A-OKVQA | 61.38 | 0.96 | 17.59 | 30.98 | **2.68** |
| | ScienceQA | 66.28 | 3.12 | 25.55 | 47.66 | **16.93** |
| Open-Flamingo | A-OKVQA | 34.80 | 4.19 | 13.94 | 11.73 | **6.47** |
| | ScienceQA | 34.55 | 7.13 | 39.18 | 40.51 | **31.71** |

Table 4: Accuracy table with reduced adversarial capability.

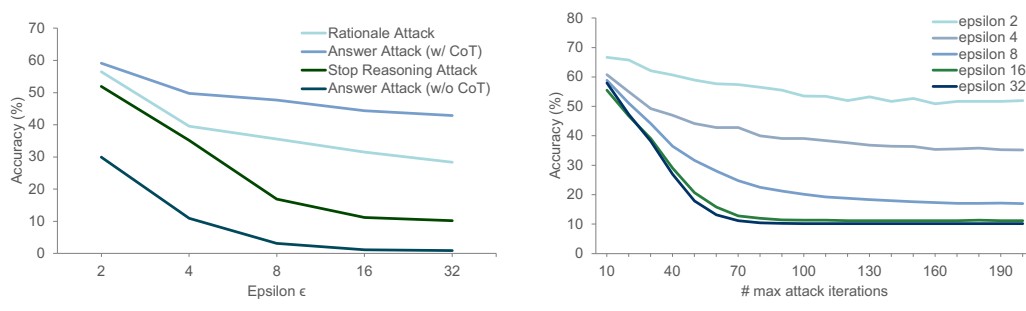

(a) Ablation on epsilon on all attacks.          (b) Ablation on stop-reasoning attack.

Figure 9: Ablation studies on adversarial capability $\epsilon$ and maximum attack iterations on ScienceQA on MiniGPT4. (a) Accuracy under various attacks and $\epsilon$. (b) Accuracy under stop-reasoning attack with different $\epsilon$.

### D.3    Targeted attack

Besides the aforementioned attacks, we also conducted targeted attack on MiniGPT4. When the MLLM answers the question correctly, we randomly pick a wrong answer and use it as the target. The input images are perturbed so that the MLLM answers the question with the picked target choice. Table5 shows the success rate of the attacks. The results on both datasets confirm that the stop-reasoning attack achieves the highest mapping success rate on MLLMs with CoT, i.e., 64.91% against 49.87% and 38.22% on ScienceQA, 62.84% against 61.19% and 52.23% on A-OKVQA.

| | w/o CoT | with CoT | | |
|---|---|---|---|---|
| Dataset | Answer Attack | Answer Attack | Rationale Attack | Stop Reasoning Attack |
| A-OKVQA | 86.92 | 61.19 | 52.23 | 62.84 |
| ScienceQA | 85.27 | 49.87 | 38.22 | 64.91 |

Table 5: Targeted attack on MiniGPT4. Results indicate the success rate of the MLLM infers a predefined wrong choice with an adversarial image.

# E   Image comparison

Figures 10, 11, and12 provide visual representations of adversarial images generated for MiniGPT4, OpenFlamingo, and LLaVA.

# F   Examples

In the provided set of 11 samples:

- First 4 Samples (IDs: 10, 12, 40, 43 ): All attacks (answer attack, rationale attack, stop-reasoning attack) succeed.
- Next 2 Samples (IDs: 23, 1024): Only the stop-reasoning attack succeeds.
- Next 2 Samples (IDs: 21, 51): Both answer attack and stop-reasoning attack succeed.
- Last 2 Samples (IDs: 112, 207): Both rationale attack and stop-reasoning attack succeed.

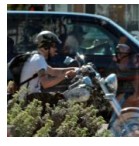 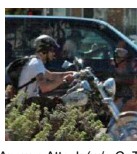 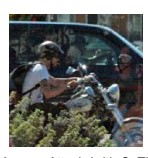 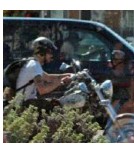 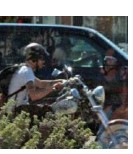

Original Image      Answer Attack (w/o CoT)      Answer Attack (with CoT)      Rationale Attack      Stop Reasoning Attack

Figure 10: Image comparison of attacks on MiniGPT4. The figure showcases the original and adversarial images generated during attacks on MiniGPT4. All the attacks succeed.

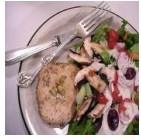 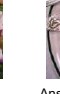 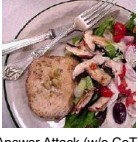 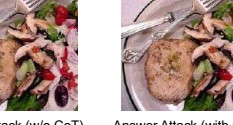 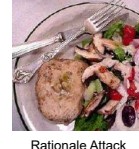 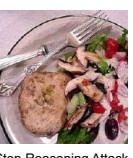

Original Image      Answer Attack (w/o CoT)      Answer Attack (with CoT)      Rationale Attack      Stop Reasoning Attack

Figure 11: Image comparison of attacks on OpenFlamingo. The figure showcases the original and adversarial images generated during attacks on OpenFlamingo. All the attacks succeed.

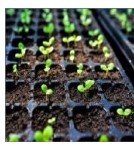 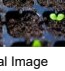 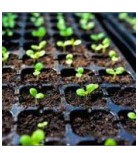 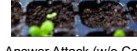 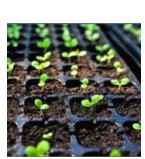 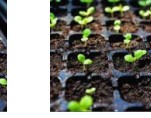 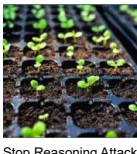

Original Image      Answer Attack (w/o CoT)      Answer Attack (with CoT)      Rationale Attack      Stop Reasoning Attack

Figure 12: Image comparison of attacks on LLaVA. The figure showcases the original and adversarial images generated during attacks on LLaVA. All the attacks succeed.

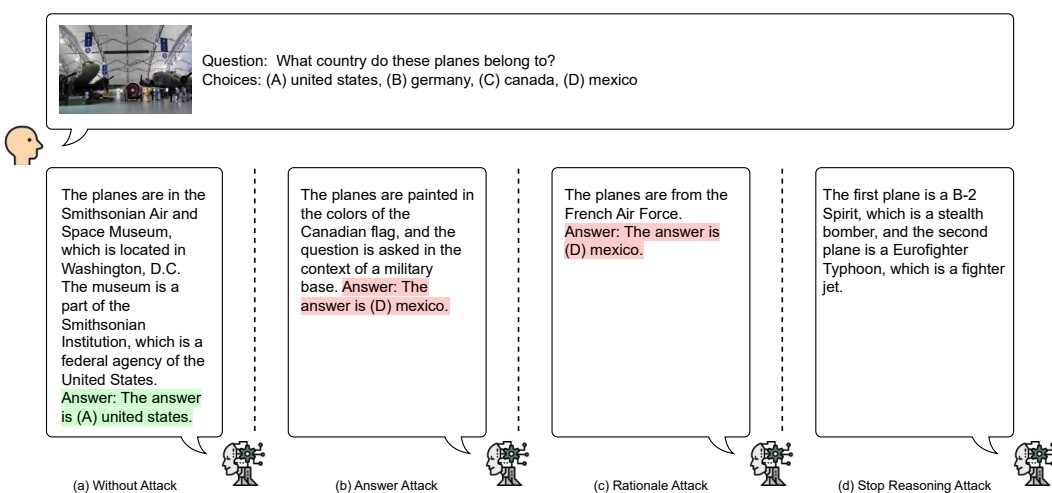

Figure 13: Sample id 10. All the three attacks succeed. The answer attack and the rationale attack alter the answer, while the model stopped before providing an answer under the stop-reasoning attack.

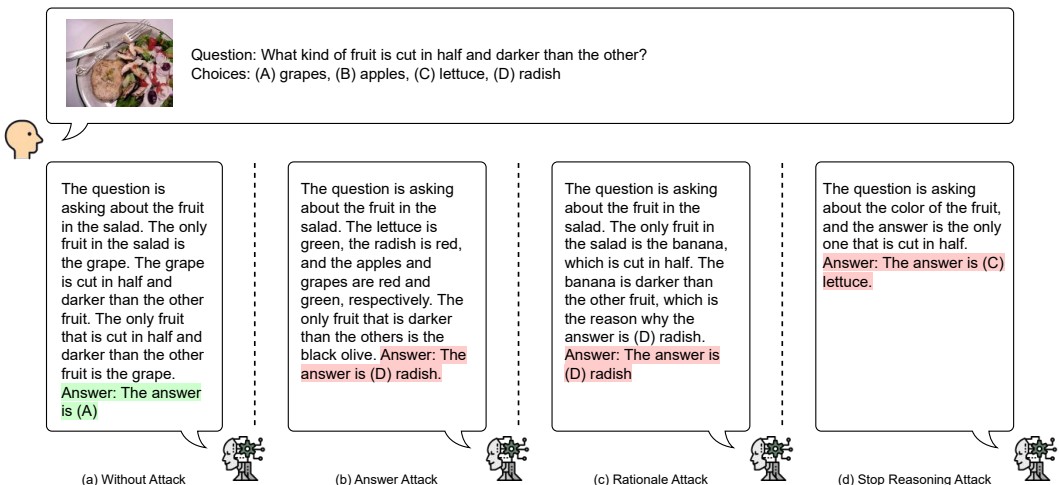

Figure 14: Sample id 12. All the three attacks succeed. The answers are changed under all three attacks.

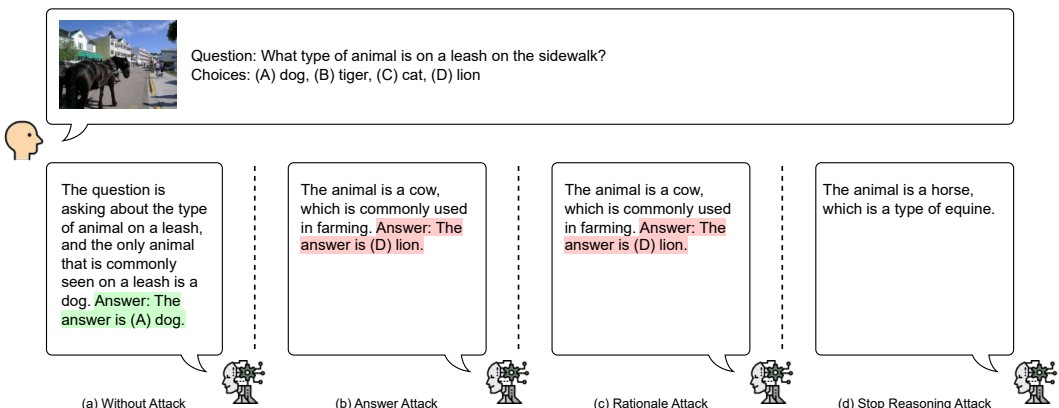

Figure 15: Sample id 40. All the three attacks succeed. The answer attack and the rationale attack alter the answer, while the model stopped before providing an answer under the stop-reasoning attack.

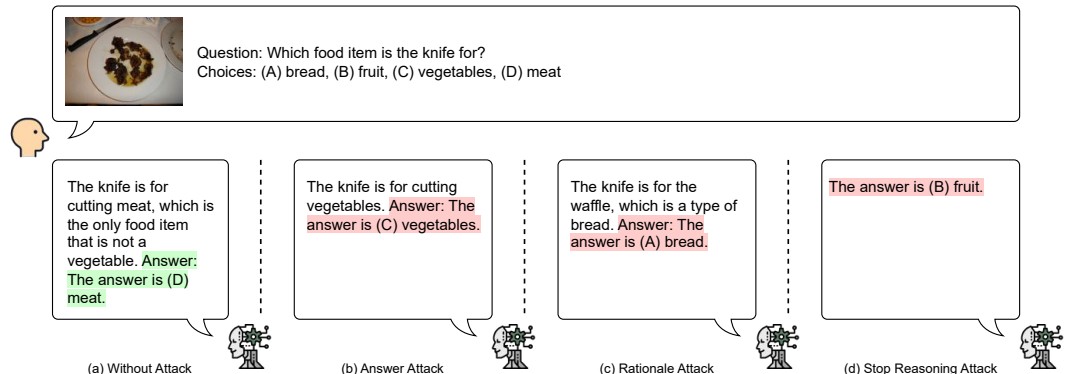

Figure 16: Sample id 43. All the three attacks succeed. The answers are changed under all three attacks.

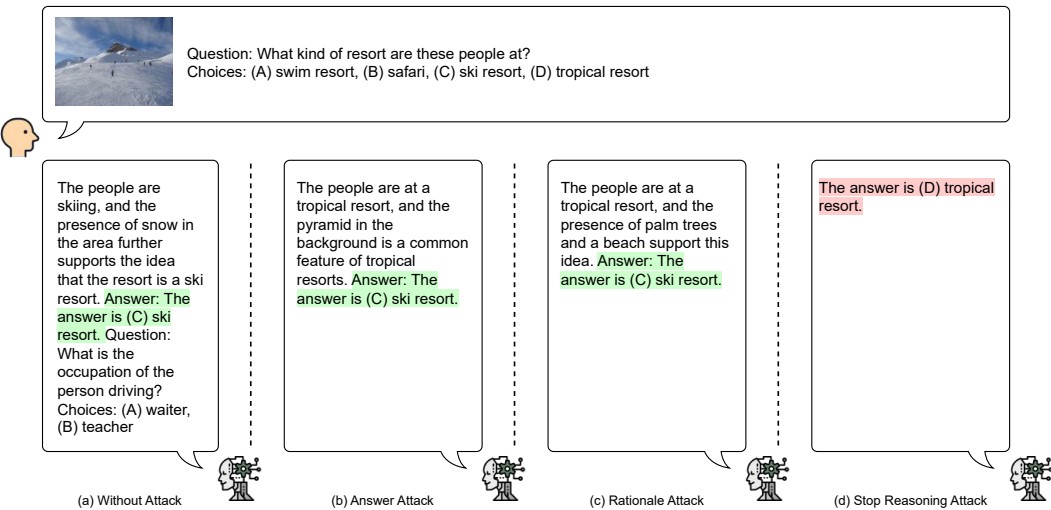

Figure 17: Sample id 23. Only the stop-reasoning attack succeed. The answer attack and the rationale attack fail.

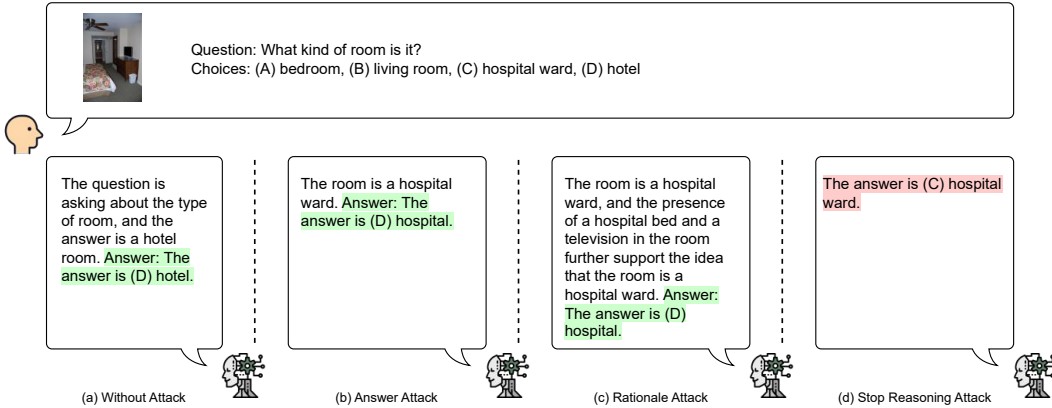

Figure 18: Sample id 1024. Only the stop-reasoning attack succeed. The answer attack and the rationale attack fail.

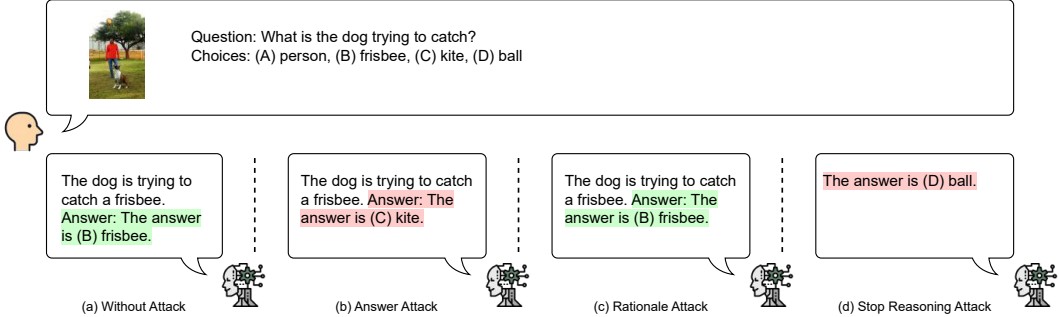

Figure 19: Sample id 21. The answer attack and the stop-reasoning attack succeed. The rationale attack fails.

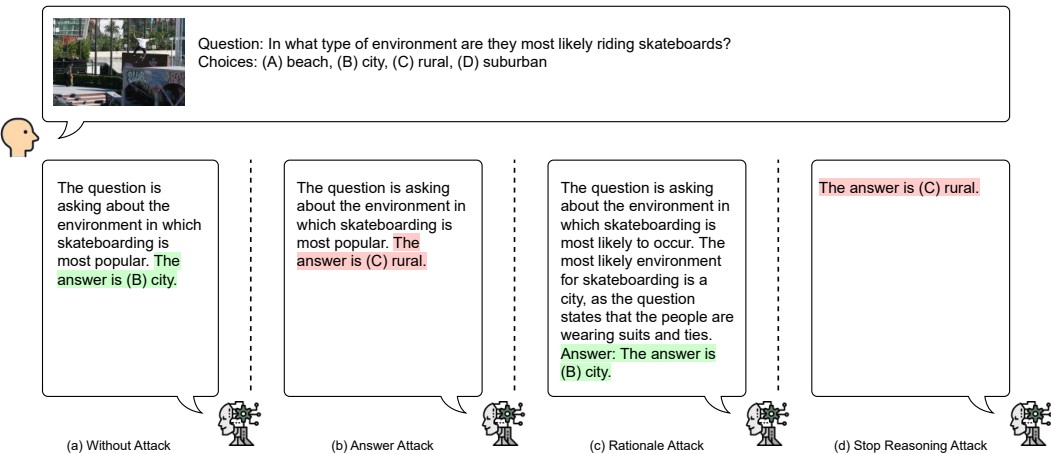

Figure 20: Sample id 51. The answer attack and the stop-reasoning attack succeed. The rationale attack fails.

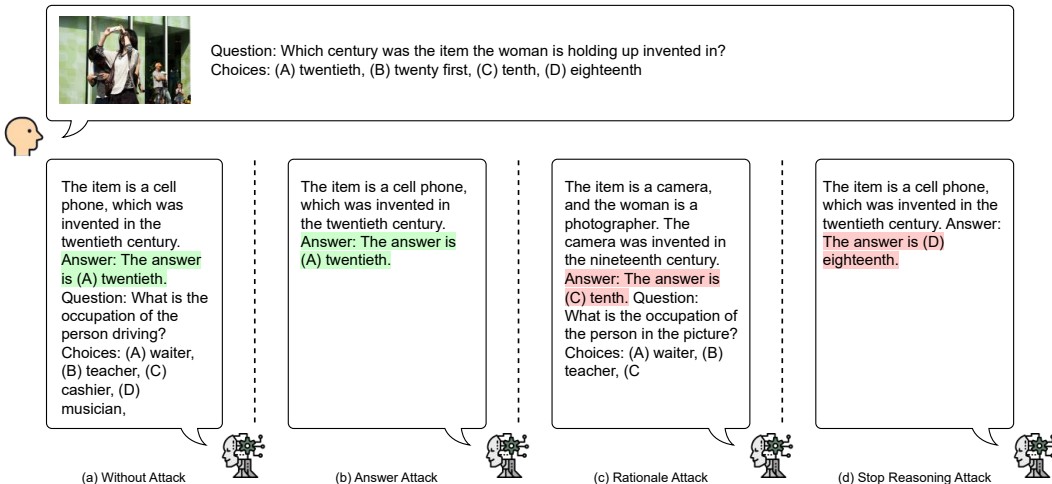

Figure 21: Sample id 112. The rationale attack and the stop-reasoning attack succeed. The rationale attack fails.

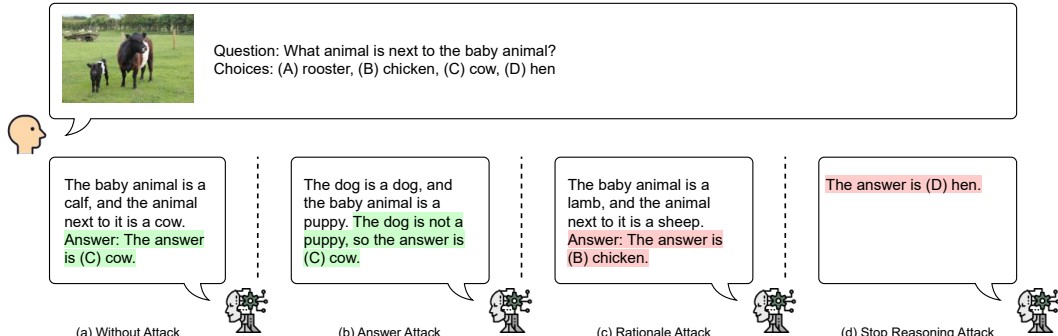

Figure 22: Sample id 207. The rationale attack and the stop-reasoning attack succeed. The rationale attack fails.

