# OpenReview forum: "Stop Reasoning! When Multimodal LLM with Chain-of-Thought Reasoning Meets Adversarial Image"
_colmweb.org/COLM/2024/Conference — COLM_

### Official Review · Reviewer_3eKb · 2024-05-10

**Rating:** 8
**Confidence:** 3
**Ethics Flag:** 1

**Summary:**

This paper considers the intersection of adversarial images + chain-of-thought prompting.

The setup involves a model answering multiple choice questions given an image. The model tries to give reasoning first before spitting out a letter from A to E. There are several attacks:
* answer attack, in making the model spit out the wrong answer
* rationale attack, in making the model point out incorrect observations
* stop reasoning attack, in making the model not do chain-of-thought at all

the authors consider various image+LLMs like MiniGPT4, OpenFlamingo, and LLaVa; then A-OKVQA and ScienceQA. the results show that these various attacks can be successful.

---
edit post rebuttal: still agree to accept this paper, I am less concerned about novelty and don't think one negative review saying that should be reason to reject. so overall should be reasonable to accept this paper given all the reviews.

**Reasons To Accept:**

This seems like an interesting paper to this reviewer. It extends the domain of adversarial inputs to the multiple-choice and CoT regime, both of which are very popular in today's LLM era. It also suggests
* CoT provides a marginal robustness boost
* Stop-reasoning is the most effective attack (so it is possible to perturb the images such that models stop reasoning)

these seem pretty useful for the community to know!

**Reasons To Reject:**

Possibly, this is a bit of a narrow attack that assumes a lot of knowledge about the model - the model's parameters, the domain, etc. To this reviewer though this is interesting even outside of the security domain, so I still recommend acceptance.

---

> ### Author Rebuttal · Authors · 2024-05-31
>
> Thank you for your positive feedback and acceptance recommendation.
>
> We are very happy that you highlighted the contributions in the paper. The results in the paper reveal that CoT marginally enhances the robustness of MLLMs, and the stop-reasoning attack, which can bypass the reasoning steps, is the most effective attack method compared with answer attack and rationale attack on MLLMs with CoT.
>
> We are thrilled to hear that our work has sparked interest beyond the security domain. We also hope that our work can inspire further research across various fields, leveraging the insights we have uncovered. We will continue our investigations in this area to contribute to the broader understanding of CoT and MLLMs' robustness and vulnerabilities.
>
> Regarding the point about the assumed knowledge of the model, the reason is that our primary focus is not solely on the robustness against adversarial images but also on how the CoT reasoning process influences MLLMs' robustness. To better understand this influence, we assumed that the adversary possesses knowledge about the MLLMs.
>
> Thank you once again for your thoughtful review and encouragement.

---

### Official Review · Reviewer_VqF6 · 2024-05-10

**Rating:** 6
**Confidence:** 4
**Ethics Flag:** 1

**Summary:**

This research focuses on multimodal language models (MLLMs) and introduces a novel adversarial attack method called "stop-reasoning attack" to investigate and assess the impact of CoT reasoning on the adversarial robustness of MLLMs. Experimental results demonstrate that CoT reasoning indeed enhances the robustness of MLLMs against existing adversarial attack methods.

**Questions To Authors:**

1. I have a question about the difference between "stop reasoning attack" and simply not using CoT and then attacking the answer. I believe they both ultimately attack the answer. What might be the performance difference between the "Answer Attack" column in Table 1 for "w/o CoT" and the "Stop Reasoning Attack" column for "with CoT"? Could this difference in performance be due to the need for additional optimization of a reasoning component in the stop reasoning attack

**Reasons To Accept:**

1. The purpose and approach of the attack are interesting.
2. The writing of the paper is also quite clear.

**Reasons To Reject:**

1. I am concerned about the novelty of the method. The overall pipeline of the attack is similar to other methods, where noise is used to perturb images so that they are mapped to incorrect target locations in the feature space after passing through the MLLM image encoder.

---

> ### Author Rebuttal · Authors · 2024-05-31
>
> Thank you for your valuable feedback.
>
> Response for "Reasons To Reject":
>
> Regarding the concern about the novelty of our method, we think there might be a misunderstanding. The primary aim of our paper is not to introduce a completely new attack method, but rather to investigate the impact of CoT reasoning on the adversarial robustness of MLLMs. We highlighted two main research questions in the introduction: (1) Does CoT enhance the adversarial robustness of MLLMs? (2) What do the intermediate reasoning steps of CoT entail under adversarial attacks?
>
> The CoT reasoning process is known to improve MLLM performance on complex reasoning tasks, but its effect on robustness has not been thoroughly explored. In our study, we conducted three different attacks on MLLMs with CoT and one attack on MLLMs without CoT. Our results indicate that while CoT provides marginal robustness enhancements, these can be negated when the model is led to skip the reasoning phase through the stop-reasoning attack. Furthermore, our analysis of the rationale changes under different attacks demonstrates how CoT serves as a window to understand why MLLMs make incorrect answers when confronted with adversarial images.
>
> Response for "Questions To Authors":
>
> Concerning your question about the performance gap between the stop-reasoning attack on MLLMs with CoT and the answer attack on MLLMs without CoT, we think this is not due to the additional optimization cost of the reasoning component. We ensured that the attack iterations were sufficiently long and the attacks converged in all experiments, ruling out optimization cost as the cause of performance differences.
>
> The difference might be caused by two factors:
> 1. The prompts for CoT and non-CoT scenarios are naturally different. We add explicit instructions such as "first, generate a rationale that can be used to infer the answer to the question" to prompt the MLLMs to leverage the CoT. In comparison, we ask MLLMs to infer the answer directly under "w/o CoT" scenarios.
> 2. The stop-reasoning attack cannot always stop MLLMs from performing reasoning. We observed that not all samples successfully skipped the reasoning phase after the stop-reasoning attack (e.g., when evaluating the stop-reasoning attack with the MiniGPT4 model on the A-OKVQA dataset, the model did not skip the reasoning phase for 5.35% samples).
>
> These two factors likely account for the performance gap between the two attacks.
>
> Thank you once again for your valuable insights.

---

> > ### Comment · Reviewer_VqF6 · 2024-06-04
> > **Official Comment by Reviewer VqF6**
> >
> > Thank you very much for your response. It has addressed my concerns and clarified a previous misunderstanding I had about this paper. I have adjusted my rating：）

---

### Official Review · Reviewer_hxRq · 2024-05-12

**Rating:** 7
**Confidence:** 4
**Ethics Flag:** 1

**Summary:**

This paper first found that the CoT technique can improve the adversarial robustness of MLLMs. Then they proposed a novel attack method that train visual noise using designed losses that are accosiated with CoT components. Experiments are conducted on two VQA datasets using three different MLLMs, and the resutls presented confim the effectiveness of the method.

**Reasons To Accept:**

1. The writing is straightforward and easy to follow.
2. The method seems to be effective given the results presented.
3. Discussion sections in the paper are insightful and the picked examples are vivid.

**Reasons To Reject:**

1. Generalization: the paper only perform their method on two VQA task. On the one hand, can the method perform well on multimodal generation tasks like captioning? On the other hand, while your current attack goal is to only "avoid the right answer", can your method be applied to more strict tasks like "generate the pre-defined wrong answer " (e.g., the jailbreaking task)?
2. Model Size: only MLLMs with 7B size LLM are applied , I'm wondering how's effectiveness of your stop-reasoning on larger MLLMs, such as models with 13B LLMs.
3. Ablation of $\epsilon$ and the training iteration: the pertubation budget $\epsilon$ and training iteration are main parameters in adversarial learning domain. Maybe you can draw a picture about the attack success rate of your method under different $\epsilon$ and/or training steps in the main part.
4. Missing reference: several references that are highly related to your paper are missing [1-3].

[1] Zhang H, Shao W, Liu H, et al. AVIBench: Towards Evaluating the Robustness of Large Vision-Language Model on Adversarial Visual-Instructions[J]. arXiv preprint arXiv:2403.09346, 2024.

[2] Tu H, Cui C, Wang Z, et al. How many unicorns are in this image? a safety evaluation benchmark for vision llms[J]. arXiv preprint arXiv:2311.16101, 2023.

[3] Liu X, Zhu Y, Lan Y, et al. Query-relevant images jailbreak large multi-modal models[J]. arXiv preprint arXiv:2311.17600, 2023.

---

> ### Author Rebuttal · Authors · 2024-05-31
>
> Thank you for your detailed feedback and constructive suggestions.
> 1. Generalization:
> - Regarding the dataset, our primary focus is to explore the influence of CoT on the adversarial robustness of MLLMs, where reasoning is necessary. The two chosen VQA datasets are representative of tasks that demand reasoning. In comparison, captioning tasks typically do not involve reasoning steps, which is why they were excluded. Moreover, since many tasks can be formulated as VQA task, our approach should remain generalizable even though we have only performed our method to two VQA tasks.
> - Concerning targeted attacks, we have conducted additional experiments on MiniGPT4 with pre-defined wrong answer. The results on both datasets confirm that the stop-reasoning attack achieves the highest mapping success rate on MLLMs with CoT, i.e., 64.91% (stop-reasoning attack) against 49.87% (answer attack) and 38.22% (rationale attack) on ScienceQA, 62.84% (stop-reasoning) against 61.19% (answer) and 52.23% (rationale) on A-OKVQA.
> 2. Model Size: We ran experiments on MiniGPT4-Vicuna13B. The results are consistent with our findings presented in the paper, demonstrating that the stop-reasoning attack is the most effective. The detailed results are presented below:
> || w/o CoT (acc%)|w/o CoT| w/ CoT (acc%)|w/ CoT|w/ CoT|
> |-|-|-|-|-|-|
> |Datasets|w/o Attack|Answer Attack|Answer Attack|Rational Attack|Stop Reasoning Attack|
> |A-OKVQA|42.65|1.45|17.97|36.84|10.53|
> |ScienceQA|63.64|13.75|33.78|45.69|30.89|
> 3. Ablation Studies: More ablations on $\epsilon$ and iteration were conducted, please see figure at [here](https://anonymous.4open.science/r/stop_reasoning_colm-BF9B). Ablation indicates that with different $\epsilon$, our stop-reasoning attack is consistently better than rationale attack and answer attack on MLLMs with CoT. Ablation on iterations shows the convergence of the different attacks.
> 4. Missing References: Thank you for pointing out the references. We will incorporate the suggested references into the related work. Specifically:
> - AVIBench (Zhang H, et al.) and MM-SafetyBench (Liu X, et al.) provide frameworks for evaluating MLLMs' robustness against adversarial attacks, but they do not focus on the CoT reasoning process.
> - Unicorn (Tu H, et al.) explores robustness against adversarial attacks and OOD generalization but does not address the influence of CoT reasoning on MLLMs' robustness.
>
> Thank you again for your valuable feedback and helpful review.

---

> > ### Comment · Reviewer_hxRq · 2024-06-05
> >
> > Thanks for your additional results, I’ll increase my score by 1.

---

### Decision · Program_Chairs · 2024-07-10

**Decision:**

Accept

**Comment:**

This paper considers the intersection of adversarial images and chain-of-thought prompting in multimodal LLMs. After author rebuttal, it received scores of 678. The authors have done a good job of rebuttal, and all the reviewers are happy about the paper after rebuttal, commenting that (1) the writing is clear and easy to follow, (2) the method seems to be effective given the results presented, and (3) it extends the domain of adversarial inputs to the multiple-choice and CoT regime in the setting of multimodal LLMs. The additional results added during rebuttal is also helpful to address reviewers' concerns. Therefore, the AC would like to recommend acceptance of the paper.